# Longitudinal kinetics of RBD+ antibodies in COVID-19 recovered patients over 14 months

Tsuf Eyran[1], Anna Vaisman-Mentesh[1], David Taussig[1], Yael Dror[1], Ligal Aizik[1], Aya Kigel[1,2], Shai Rosenstein[1], Yael Bahar[1], Dor Ini[1], Ran Tur-Kaspa[3,4,5], Tatyana Kournos[6], Dana Marcoviciu[6], Dror Dicker[6,7]*, Yariv Wine[1,2]*

1 The Shmunis School of Biomedicine and Cancer Research, George S. Wise Faculty of Life Sciences, Tel Aviv University, Tel Aviv, Israel, 2 The Center for Combatting Pandemics, Tel Aviv University, Tel Aviv, Israel, 3 Liver Institute, Rabin Medical Center, Beilinson Hospital, Petah Tikva, Israel, 4 Molecular Hepatology Research Laboratory, Felsenstein Medical Research Center, Sackler School of Medicine Tel-Aviv University, Tel Aviv, Israel, 5 The Azrieli Faculty of Medicine, Bar-Ilan University, Safed, Israel, 6 Internal Medicine D, Hasharon Hospital, Rabin Medical Center, Petah Tikva, Israel, 7 Sackler School of Medicine, Tel Aviv University, Tel Aviv, Israel

☯ These authors contributed equally to this work.
* daniel3@013.net (DD); YarivWine@tauex.tau.ac.il (YW)

**Data Availability Statement:** For ethical and legal reasons, the data is not publicly available. The data which supports the findings of this study can be accessed by interested researchers from Tel Aviv

## Abstract

We describe the longitudinal kinetics of the serological response in COVID-19 recovered patients over a period of 14 months. The antibody kinetics in a cohort of 192 recovered patients, including 66 patients for whom follow-up serum samples were obtained at two to four clinic visits, revealed that RBD-specific antibodies decayed over the 14 months following the onset of symptoms. The decay rate was associated with the robustness of the response in that antibody levels that were initially highly elevated after the onset of symptoms subsequently decayed more rapidly. An exploration of the differences in the longitudinal kinetics between recovered patients and naïve vaccinees who had received two doses of the BNT162b2 vaccine showed a significantly faster decay in the naïve vaccinees, indicating that serological memory following natural infection is more robust than that following to vaccination. Our data highlighting the differences between serological memory induced by natural infection vs. vaccination contributed to the decision-making process in Israel regarding the necessity for a third vaccination dose.

## Author summary

The fundamental idea guiding vaccine science is that an ideal vaccine should induce immunity similar to the immunity produced by natural infection. A vaccine is designed to "train" the immune system in a way that it will mimic the stimulation necessary for immune development, yet not produce active disease. Understanding the persistence of antibodies in patients following recovery from natural infection with SARS-CoV-2 will help to highlight the differences between the breadth of the antibody responses following natural infection and vaccination and may inform us whether the vaccine "training" will effectively stimulate the immune system to provide long-lasting immunity. Using samples

University. To access the data, please contact Mrs. Tal Oded, senior administrative assistant, The Shmunis School of Biomedicine and Cancer Research, George S. Wise Faculty of Life Sciences, Tel Aviv University, Israel, Email: taloded@tauex. tau.ac.il.

**Funding:** The study was supported by the grant #3-17162 from the Israeli Ministry of Health (Y. W.). The funders had no role in study design, data collection and analysis, decision to publish, or preparation of the manuscript.

**Competing interests:** The authors have declared that no competing interests exist.

collected from recovered COVID-19 patients over an extended period of 14 months, we followed the persistence of antibodies and found an association between the antibody levels in proximity to recovery and the rate of decay. In addition, we found that the decay rate of antibodies in BNT162b2 vaccinees was significantly faster than that in recovered patients, suggesting that there are fundamental differences between the mechanisms of activation of the adaptive arm of the immune response following vaccine and natural infection. While natural infection involves full systemic activation, this activation may be incomplete with an mRNA vaccination, thereby affecting the capacity of the immune system to maintain an antibody reservoir over time.

## Introduction

The first patients with coronavirus disease 2019 (COVID-19) were identified in Wuhan, China in December 2019 [1]. These patients were found to be infected by severe acute respiratory syndrome coronavirus 2 (SARS-CoV-2), and their identification was followed some weeks later by a declaration of the World Health Organization (WHO) that COVID-19 had become a worldwide pandemic [2,3]. Rapid response to the outbreak provided important information regarding the virus genome sequence and especially the spike protein (S protein) and its subregion, the receptor binding domain (RBD) that is responsible for binding to human angiotensin-converting enzyme 2 (hACE2) to mediate virus entry into the cells [4]. The ability to measure antibody responses to SARS-CoV-2 antigens is vital for ascertaining past viral exposure, investigating transmission in the community, and carrying out serosurveys [5]. Thus, for evaluating the persistence of serological memory, it is essential to obtain information on the longitudinal kinetics of the antibody immune response following COVID-19 recovery.

In addition to the S protein, SARS-CoV-2 possesses three other structural proteins, namely, the membrane (M), envelope (E) and nucleocapsid (N) proteins, with the initial thinking being that the S and N proteins would be the best candidates for use as targets for measuring antibody levels [6]. Nevertheless, it was shown that, by virtue of its high immunogenicity, the viral S protein is more suitable for this purpose than the viral N protein. Moreover, the S protein RBD has the potential to elicit neutralizing antibodies that block the interaction of the virus with the host receptor hACE2 [7–9], thereby leading to viral neutralization, and, as such, it can be used as a marker for functional immune responses [10]. The utility of RBD as a marker is further supported by the high correlation between RBD-specific (RBD$^+$) antibody levels and antibody neutralization capacity [11,12]. Efforts have therefore been made to follow the kinetics of SARS-CoV-2 antibodies over the acute course of the disease and following recovery. The S protein RBD elicits antibodies starting as early as 5–15 days following the onset of symptoms [13,14], with antibody levels increasing with the progression of the disease [15–17].

Several months after the beginning of the pandemic, studies began to appear reporting that antibodies in COVID-19 recovered patients decay over time. Specifically, it was estimated that the half-life of SARS-CoV-2 anti-spike antibodies can persist for > 7 months [18], in some cases > 10 [19–21], or 13 months [22]. A recent study that examined long-lived plasma cells in the bone marrow indicated that SARS-CoV-2 infection induces a robust antigen-specific, long-lived humoral immune response in humans [23]. However, the longitudinal kinetics of viral-specific antibodies in COVID-19 recovered patients has not been clarified definitively, and reports regarding the persistence of anti-SARS-CoV-2 antibodies are inconsistent [24–29]. For example, Ortega et al. reported that the levels of spike-specific IgG/A/M levels in a

cohort of 578 COVID-19 recovered patients were stable over a period of 6 months. They attributed the antibody persistence to pre-existing immunity induced by human coronaviruses causing the common cold [29]. In a contradictory study, Isho et al. revealed that at three months following the onset of COIVD-19 symptoms, S-protein specific IgA and IgM antibodies exhibited rapid decay, while IgG antibodies remained relatively stable over time [30]. Other studies have generally agreed with the findings of Isho et al. that all antibody isotypes do, in fact, decay [28,31,32]. Gallais et al. and Pelleau et al., for example, demonstrated how studies conducted over a period of the order of 12 months on large cohorts showed a clear decay rate of all antibody isotypes [22,33]. However, the various studies have reached different conclusions regarding the exact rate of the decay. We posit that this inconsistency derives from longitudinal span restrictions (under 4 months).

Due to the high heterogenicity of the antibody response in COVID-19 recovered patients, it would appear that the best way to address the challenge of determining the accurate longitudinal kinetics of the response would be at the level of individuals who exhibit a similar antibody response, as measured in proximity to recovery. Indeed, it has previously been demonstrated that stratifying patient cohorts into sub-groups based either on the antibody levels that patients exhibited in proximity to recovery or on the antibody dynamics profile provide more accurate measures of the antibody kinetics [34,35]. Nevertheless, it still remains necessary to determine the decay rate in a more accurate manner that considers the heterogenicity of the antibody response in recovered patients. Therefore, we conducted a longitudinal study of changes in RBD$^+$ antibody levels in samples obtained from 192 COVID-19 recovered patients over a period spanning up to 450 days following the onset of symptoms. Out of the 192 patients, follow-up samples from 66 COVID-19 recovered patients were collected across 4 clinic visits (designated V1-V4), with an interval of approximately 90 days between visits. IgG, IgM and IgA were determined, and kinetic parameters were calculated. Overall, our data provides new insights into the longitudinal antibody kinetics of COVID-19 recovered patients that may contribute to decision making regarding vaccine design and vaccination regimes.

# Results

## Cohort establishment and antibody measurement

To investigate the longitudinal kinetics and persistence of SARS-CoV-2 specific antibodies, we studied a cohort of 192 COVID-19 patients who had recovered from SARS-CoV-2 infection. COVID-19 was diagnosed on the basis of a positive qPCR result for SARS-CoV-2, and the metric of days following the onset of symptoms (DFS) was set according to self-reporting. The characteristics of the cohort were as follows: median age 53 years (range 20 to 81 years), balanced gender distribution of males and females (M:F 50.8%:49.2%), and disease severity mild in 83% of cases and moderate/severe in 17%. Blood samples were collected from all participants at visit 1 (V1, mean 90 DFS), with follow up collections at visit 2 (V2) from 34.4% participants (mean 184 DFS), at visit 3 (V3) from 12.5% participants (mean 298 DFS) and at visit 4 (V4) from 2.6% participants (mean 406 DFS) (Figs 1A and 1B and S1A, and S1 Table).

A sub-cohort of the COVID-19 recovered patients comprised 18 patients who were vaccinated with one dose of the BNT162b2 COVID-19 mRNA vaccine within a time window of approximately 222 DFS. Also included in the study was a prospective cohort of 17 naïve individuals who received two doses of the BNT162b2 mRNA-vaccine with an interval of 21 days between doses, according to the official guidelines of the Israel Ministry of Health (MOH). Samples from this cohort were collected at 8 days following vaccination after the first dose (DFV$^{x1}$) and at four visits following the second vaccine dose, spaced at a median of 8, 35, 91, and 182 days following vaccine (DFV$^{x2}$). RBD$^+$ IgG, IgM and IgA levels were measured by

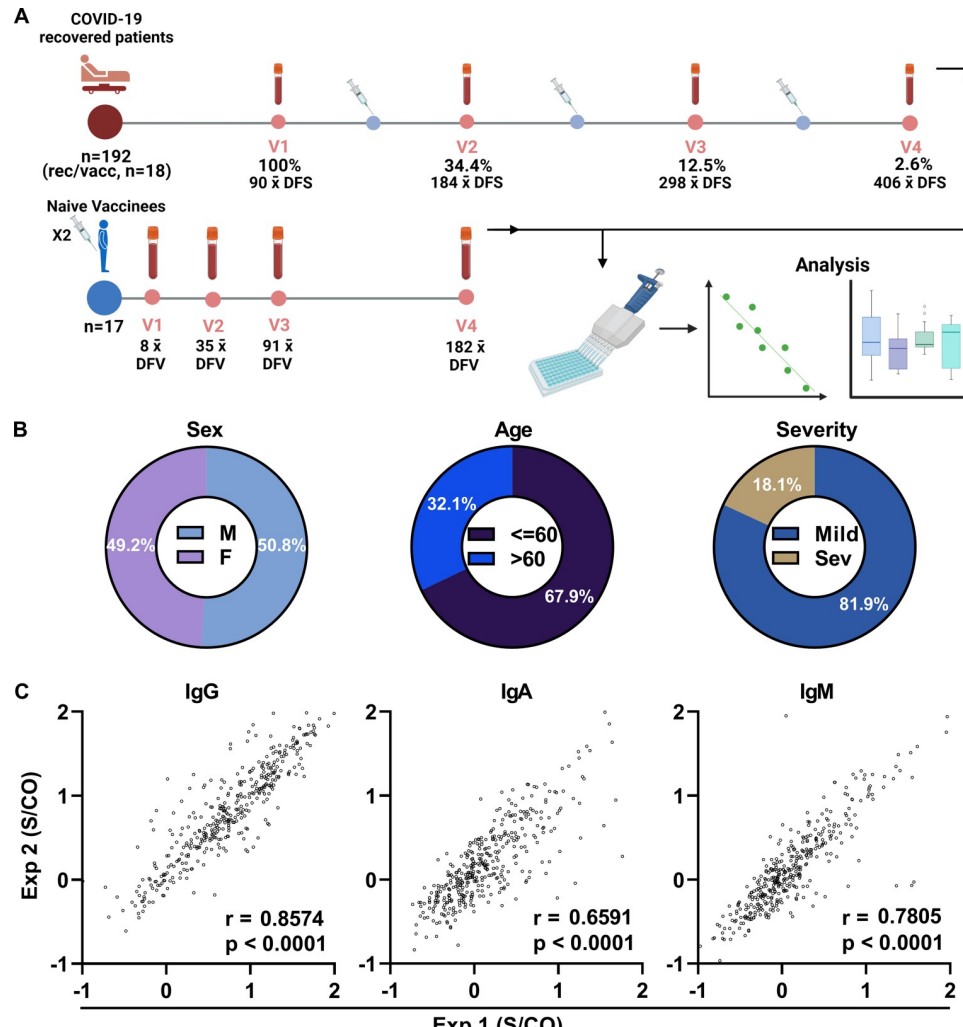

**Fig 1. Experimental design, cohort characteristics, and reproducibility of antibody level measurements.** (A) The cohort comprised 192 COVID-19 recovered patients; blood samples were obtained from all the patients at V1, and from 66 patients at subsequent visits (V2-V4) at intervals of approximately 3 months between visits. A subset of the patients (n = 18) received one dose of an mRNA-vaccine at various time points [days following the onset of symptoms (DFS)], indicated by syringe icons at various time points. Blood samples were also collected from a cohort of naïve vaccinees (n = 17) who received 1 and 2 doses the mRNA-vaccine. All blood samples were fractionated for the isolation of plasma samples, which were used for the determination of the RBD+ antibody levels and for the analysis of the longitudinal kinetics. (B) The characteristics of the cohort of COVID-19 recovered patients included a balanced male/female population, with 67.9% below and 32.1% above 60 years. The majority of patients exhibited mild symptoms (81.9%), but symptoms were severe in 18.1% of patients. (C) RBD+ measurements were carried out in duplicate, and each experiment was repeated twice (designated Exp1 and Exp2 on the x- and y-axes, respectively). Values on the x and y axes are presented as $\log_{10}$ signal over cutoff (S/CO). Pearson's correlation coefficient (r) was used to determine the reproducibility of the experiments. p values < 0.05 were considered significant.

semi-quantitative ELISA, and antibody levels were normalized by calculating the ratio of the signal to the mean signal obtained in the negative controls used in each microwell plate (signal over cutoff, S/CO). Each sample was measured in duplicate, and each experiment repeated independently twice. Technical duplicate measurements demonstrated high reproducibility, as determined by Pearson's correlation coefficients of r = 0.8574, 0.6591, 0.7805 for IgG/A/M, respectively (Fig 1C). In total, 287 samples were collected from COVID-19 recovered patients

with 192 at V1, and an additional 95 follow-up samples were obtained at V2-V4. Sixty-one samples were collected (n = 18) from the sub-cohort of recovered and vaccinated individuals, and 68 samples, from 17 naïve vaccinees collected during 4 visits (S1B Fig).

## Association of antibody levels with age, sex, and disease severity

To test whether the robustness of the antibody response following SARS-CoV-2 infection is associated with patients' age and sex and the severity of the COVID-19 disease, RBD$^+$ antibody levels at V1 were stratified according to these three factors. We found that the IgG and IgA, but not IgM, levels differed significantly between male and female patients (Fig 2A). When the cohort was stratified by age >60 and ≤ 60 years, levels of RBD$^+$ IgG and IgA isotypes were significantly higher among the >60 group, while IgM was found not to be significantly different (Fig 2B). Disease severity during the active phase of the disease was associated with higher IgG and IgA levels at V1, and there was no association between IgM and disease severity (Fig 2C).

## Decay in antibody levels over the study period of 14 months

The serum samples that were collected from the COVID-19 recovered patients (n = 287) were used to evaluate the persistence of the RBD$^+$ antibodies over the study period of 14 months. The association between the time of onset of symptoms (in DFS) and RBD$^+$ IgG/A/M levels was examined by applying a linear regression model, and the extent of change over time was determined by the regression coefficient (i.e., the slope of the regression). For all antibody isotypes, the RBD$^+$ levels were found to decay over the 14-month period (S2 Fig), as was indicated by the negative values of the regression coefficient. The regression coefficients differed significantly between the isotypes, with IgA showing the significantly fastest decay, followed by IgM and IgG (S2 Fig). Based on the linear regression model, IgG, IgA, and IgM RBD$^+$ antibody levels decayed over time in 18.4%, 61.81%, and 54.86% of serum samples, respectively, falling to the antibody levels determined for the negative control range within the time frame tested. However, upon evaluating the fitness of the regression model, we found that due to the high heterogenicity of the antibody levels among the patients, the regression model did not fit well. Thus, to better describe the antibody longitudinal kinetics, we applied the generalized additive mixed model

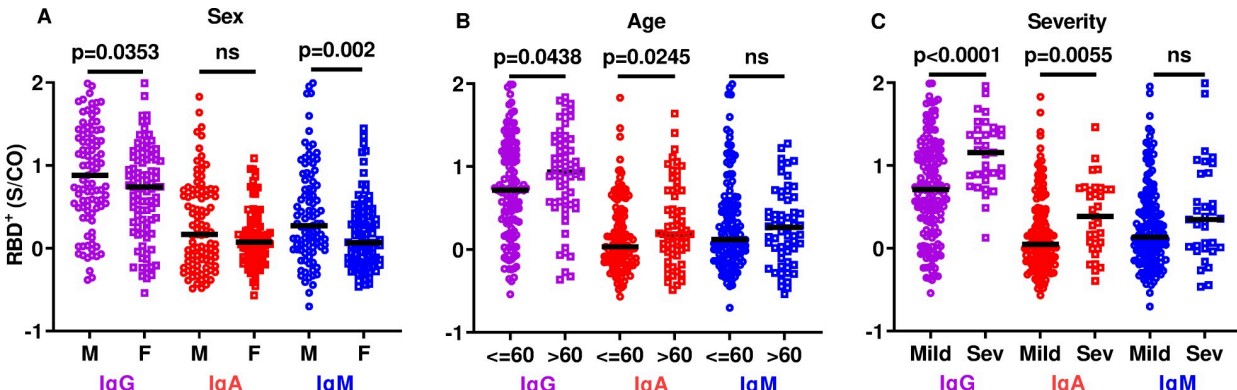

**Fig 2. SARS-CoV-2 RBD$^+$ antibody levels at V1 by sex, age, and disease severity.** Scatter plots of RBD$^+$ IgG/A/M levels in COVID-19 recovered patients at V1, by sex (A), age (B), and COVID-19 severity (C). Values on the y-axis are presented as the ratio between the signal obtained to the mean signal for the negative controls (cutoff) in each microwell plate (S/CO). Mean values are indicated by solid black lines. P values were determined using an unpaired, two-sided Mann-Whitney U-test; p < 0.05 was considered statistically significant. M—male; F—female; Mild—mild symptoms during the active phase of the disease; Sev—severe symptoms during the active phase of the disease.

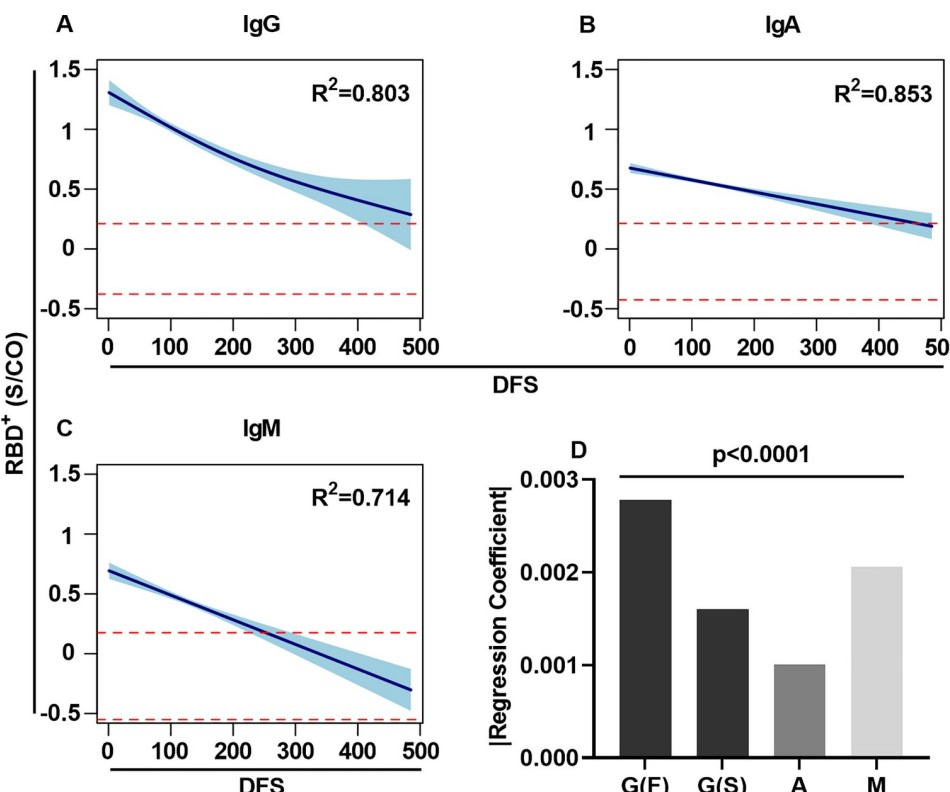

**Fig 3. RBD+ antibody longitudinal kinetics over the period of 14 months.** GAMM analysis of RBD+ antibody longitudinal kinetics for IgG (A), IgA (B) and IgM (C). The y-axis in $\log_{10}$ and the x-axis show days following symptoms (DFS). Dashed red lines represent the maximum and minimum range of the antibody levels, as measured in negative control samples (n = 27). GAMM analysis was used to determine the regression coefficient for each isotype. Negative regression coefficients (i.e., slope) of a two-phase decay or a linear decay were determined for IgG, IgA, and IgM, indicating that all isotypes decayed over the 14-month period. The 95% CI is plotted in turquoise. (D) Bar plot showing the multiple comparison analysis of the regression coefficients obtained for IgG in its fast (F) and slow (S) decay rate phases and the decay rate of IgA, and IgM. For all graphs, p values, $R^2$, effective degrees of freedom (edf) and slopes are shown in S2 Table. p values < 0.05 were considered significant.

(GAMM). GAMM analysis provided an improved regression model to describe the longitudinal kinetics of RBD+ antibodies, as evaluated by $R^2$ (Fig 3A–3C). This regression model describing the longitudinal kinetics of RBD+ IgG exhibited a two-phase decay profile, while IgA and IgM exhibited a near linear regression profile [as evaluated by the effective degree of freedom (edf) of the fitted model]. The initial decay rate for IgG was rapid, but by 200 DFS this decay had slowed down. Thus, we determined the regression coefficients for IgG utilizing two linear regression phases, one for the fast (F) decay rate and the other for the slow (S) rate. As the regression model for IgA and IgM (which decayed linearly) did not fit a two-phase decay model, the decay rate was determined by using a linear regression mixed model (as determined by GAMM analysis). Overall, the difference between the decay rates was determined by multiple comparison of the regression coefficients, which revealed that the IgG decay rate in both the F and S phases was significantly faster than the decay of IgA [IgG(F) > IgG(S) > IgA]. The decay rate of IgG in the fast phase was significantly faster than the decay of IgM, and this trend was reversed 200 DFS [IgG (F) > IgM > IgG(S)] (Fig 3D).

## Longitudinal kinetics of RBD$^+$ antibodies are associated with the robustness of the antibody response

The longitudinal kinetics of RBD$^+$ antibodies at the individual level was determined by measuring RBD$^+$ antibody titers in follow-up samples from 66 COVID-19 recovered patients. There was a marked decay of antibody levels in all follow-up patients. At V1, the fraction of patients who exhibited antibody levels that were within the range of the negative control was 19.8%, 61.4%, and 50.5% for IgG, IgA, and IgM, respectively. At the last visit, the fraction of patients (among the recurring patients) who exhibited antibody levels within the range of the negative control was 27.3%, 77.3%, and 74.2% for IgG, IgA, and IgM, respectively (Fig 4A). Due to the heterogenicity of antibody titers on an individual patient level at V1 for all isotypes (S3A Fig), we deemed it necessary to compare the longitudinal kinetics among groups of patients who had exhibited similar antibody levels at V1. To do so, we stratified patients on the basis of RBD$^+$ antibody levels into quartiles. The quartiles were determined by bootstrapping the RBD$^+$ antibody levels obtained from all serum samples (n = 287), and quartile thresholds were set according to the mean values derived from each bootstrap iteration (S3B Fig). The threshold values were then used to assign each patient to the relevant quartile according to the antibody titers at V1 (i.e., from low to high Q1-Q4, S3C Fig). Follow-up samples from each patient were used to determine the regression coefficient for each quartile group by using GAMM analysis. For example, patients for whom RBD$^+$ antibody titers at V1 were above the Q4 threshold were used to determine the regression coefficient based on those patients' follow-up samples obtained at subsequent visits. It was found that for all isotypes, patients that were assigned to quartile Q4, Q3, or Q2 exhibited negative regression coefficient values, indicating that their antibody levels had declined over time (Figs 4B and S4).

All patients who exhibited antibody titers in the range of the negative control were stratified to Q1 and were thus excluded from downstream analysis. Multiple comparison analysis of the decay rate between isotypes and quartiles was facilitated by determining the linear regression of the decay phases, where the regression was fitted to a two-phase model. A comparison of the linear regression coefficients indicated that the decay rate was significantly faster for those patients whose antibody levels at V1 had stratified them into Q4 compared to the decay in quartiles Q3 and Q2 [i.e., Q4/IgG (F) > Q3/IgG (F) > Q2/IgG (F), Fig 5]. The only exceptions to this general trend were found for IgA in the slow phase and for IgM in Q2 and Q4, i.e., Q4/IgA (S) < Q3/IgA (S) and Q2/IgM < Q4/IgM (S).

## Levels of RBD$^+$ antibodies in COVID-19 recovered patients decay more slowly than those in naïve vaccinees

To examine whether the longitudinal antibody kinetics following recovery differ from the kinetics following vaccination, we measured RBD$^+$ IgG/A/M titers in samples collected at four visits from 17 naïve individuals who received 2 doses of the mRNA BNT162b2 vaccine. Samples were obtained, on average, 8 DFV$^{x2}$, and follow-up samples on (average) 35, 91 and 182 DFV$^{x2}$. Additional samples were obtained from the same cohort, 8 DFV$^{x1}$ (1$^{st}$ dose). Based on the quartile stratification approach described above, the RBD$^+$ IgG levels at 8 DFV$^{x2}$ in all 17 naïve vaccinees were assigned to the highest quartile (Q4). This was not the case for IgA and IgM, which were distributed across the quartiles (Figs 6A and S5). The rate of antibody decay during the first 90 days post vaccination, as determined by GAMM analysis, revealed that IgA exhibited the fastest decay rate, followed by IgG and IgM (Fig 6B). We note here that we compared the decay rate only in the 90-DFV$^{x2}$ time frame, because the IgA/M decay model reached the negative range rapidly. The longitudinal kinetics for IgG exhibited a linear decay as opposed to the two-phase decay that was observed in the COVID-19 recovered patients.

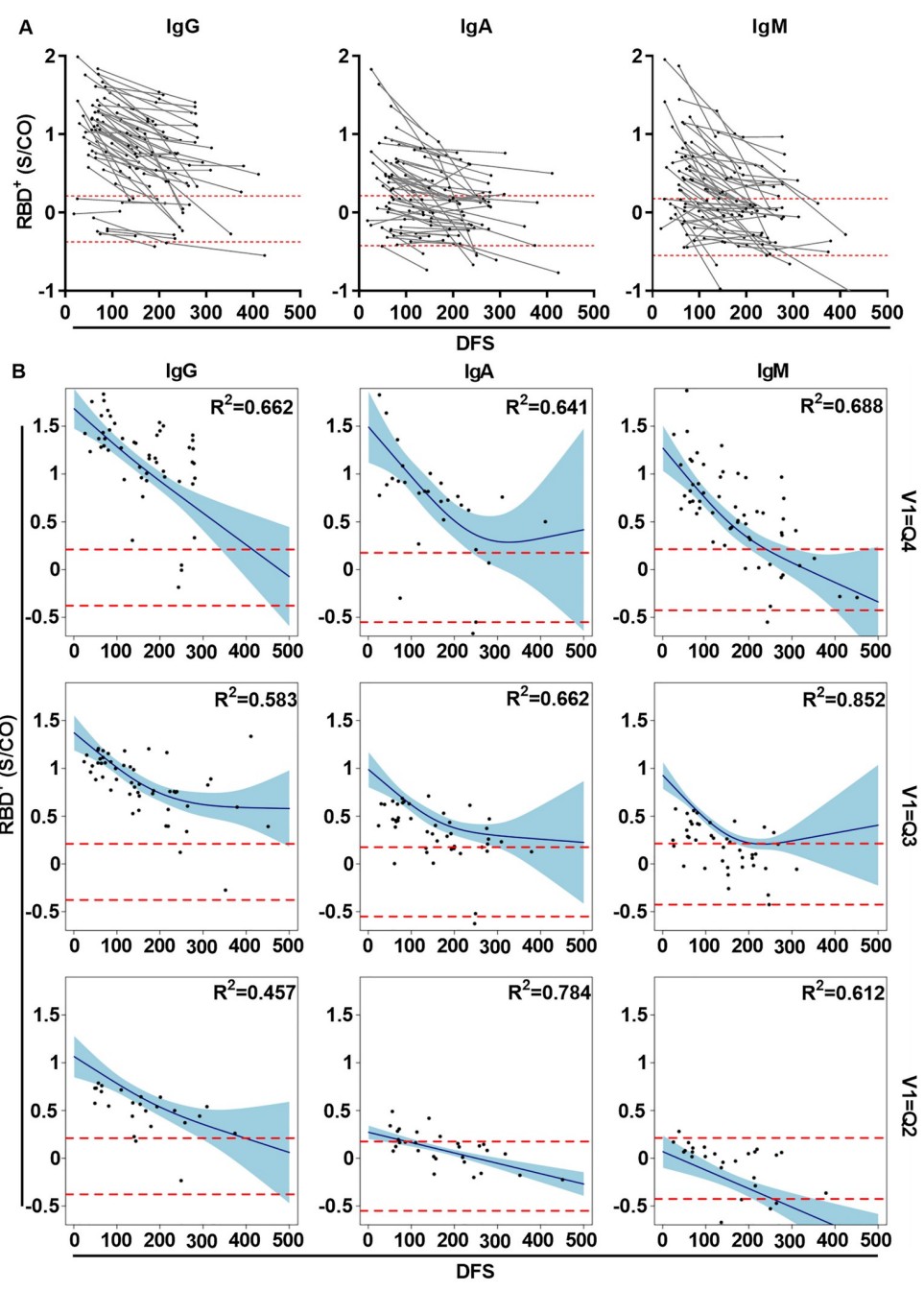

**Fig 4. Longitudinal kinetics of RBD$^+$ antibodies in COVID-19 recovered follow-up patients.** (A) RBD$^+$ IgG/A/M levels in COVID-19 recovered patients for whom we collected follow-up samples at visits 2–4. Y-axis in $\log_{10}$ and x-axis show days following symptoms (DFS). Dashed red lines represent the range of the antibody levels as measured in negative control samples (n = 27). Follow up samples from the same patient are connected by gray lines. (B) GAMM analysis of RBD$^+$ antibody longitudinal kinetics stratified by quartiles. The antibody levels at V1 determined the assigned quartile, and follow-up samples from the same patients were used to determine the regression coefficients for each quartile. Four quartiles for each isotype were analyzed. The 95% CI is plotted in turquoise. For all plots, p values, $R^2$, effective degrees of freedom (edf), and regression coefficients are shown in S2 Table. p values < 0.05 were considered significant.

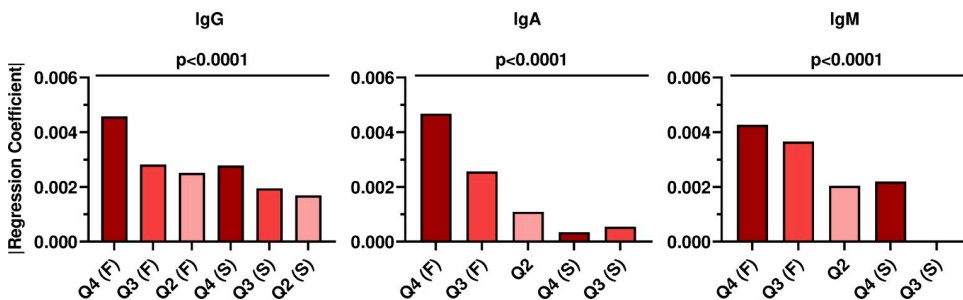

**Fig 5. Comparison of RBD+ antibody decay between quartile groups for each isotype in COVID-19 recovered patients.** Linear regression models were fitted to each quartile for all the isotypes. The linear regression coefficients were determined from the F and S phases of the two-phase decay model. Y-axis shows the absolute regression coefficient, x-axis represents the quartile and the phase of the decay (F/S and no indication for F/S if the regression did not fir a two-phase decay model) For all plots, p values, $R^2$, effective degrees of freedom(edf), and slopes are shown in S2 Table. p values < 0.05 were considered significant.

Interestingly, the RBD+ IgA titer at 8 $DFV^{x2}$ was relatively low and fell into the negative range within 50 $DFV^{x2}$. The IgM titer was within the negative range from the start, indicating that the vaccine does not elicit a strong IgM response. Since the RBD+ IgG titers at V1 in naïve vaccinees were assigned to Q4, we compared the regression coefficient for IgG for this group with that in the recovered patient group that was assigned to Q4 at V1. Using GAMM and multiple comparison analysis, we found that the decay rate in COVID-19 recovered patients was significantly slower than that in naïve vaccinees (Fig 6C and 6D). Moreover, we found that the IgG levels in 100% of the naïve vaccinees reached the negative range at 182 $DFV^{x2}$. In contrast, for only 5% of the recovered patients, IgG titers reached the negative control range at 182 DFS and for 19% of these patients, at 265 DFS (average time of their V4).

In addition, antibody levels following one vaccination dose in recovered patients (named recovered/vaccinated) were evaluated in a subset of the cohort (n = 18). As sample collection was not carried out at the day of vaccination or in proximity to that time point (0 days following vaccine, 0 $DFV^{x1}$), there was no information regarding the antibody levels at 0 $DFV^{x1}$. To fill this information gap and to enable an accurate description of the antibody kinetics in this sub-cohort, we extrapolated the antibody levels from previous visits for each individual to 0 $DFV^{x1}$ based on the regression model derived from the antibody kinetics. Likewise, to address the lack of information at 8 $DFV^{x1}$ for the recovered/vaccinated patients, we used a similar extrapolation procedure exploiting regression model derived from previous visits (V2-V3). This approach was first exemplified in one patient, Patient S106 (S6 Fig). That patient had given blood samples at three visits (V1-V3) prior to vaccination. At 366 DFS, one dose of mRNA vaccine was administered to this recovered patient. A blood sample was not obtained at the time, but a blood sample was obtained at the subsequent clinic visit (V4). Since blood samples had not been drawn from this patient on 0 $DFV^{x1}$ and 8 $DFV^{x1}$, any description of the kinetics based solely on the available samples (obtained at V3 and V4) is likely to be misleading, because it would give the impression that there had been a continuous increase in antibody levels from V3 to V4 (S6A Fig). To correct this biased representation, we extrapolated the 0 and 8 $DFV^{x1}$ from the regression model of the existing time points, thereby enabling us to give a more accurate description of the antibody longitudinal kinetics (S6B Fig). We applied this approach on all recovered/vaccinated individuals (Fig 6E). Based on the longitudinal kinetics as observed in the recovered/vaccinated sub-cohort, we found that 83% of the recovered patients who had received a single dose of the vaccine were assigned to Q4. In contrast, only 50% of the same group of individuals had been assigned to Q4 at V1 following recovery

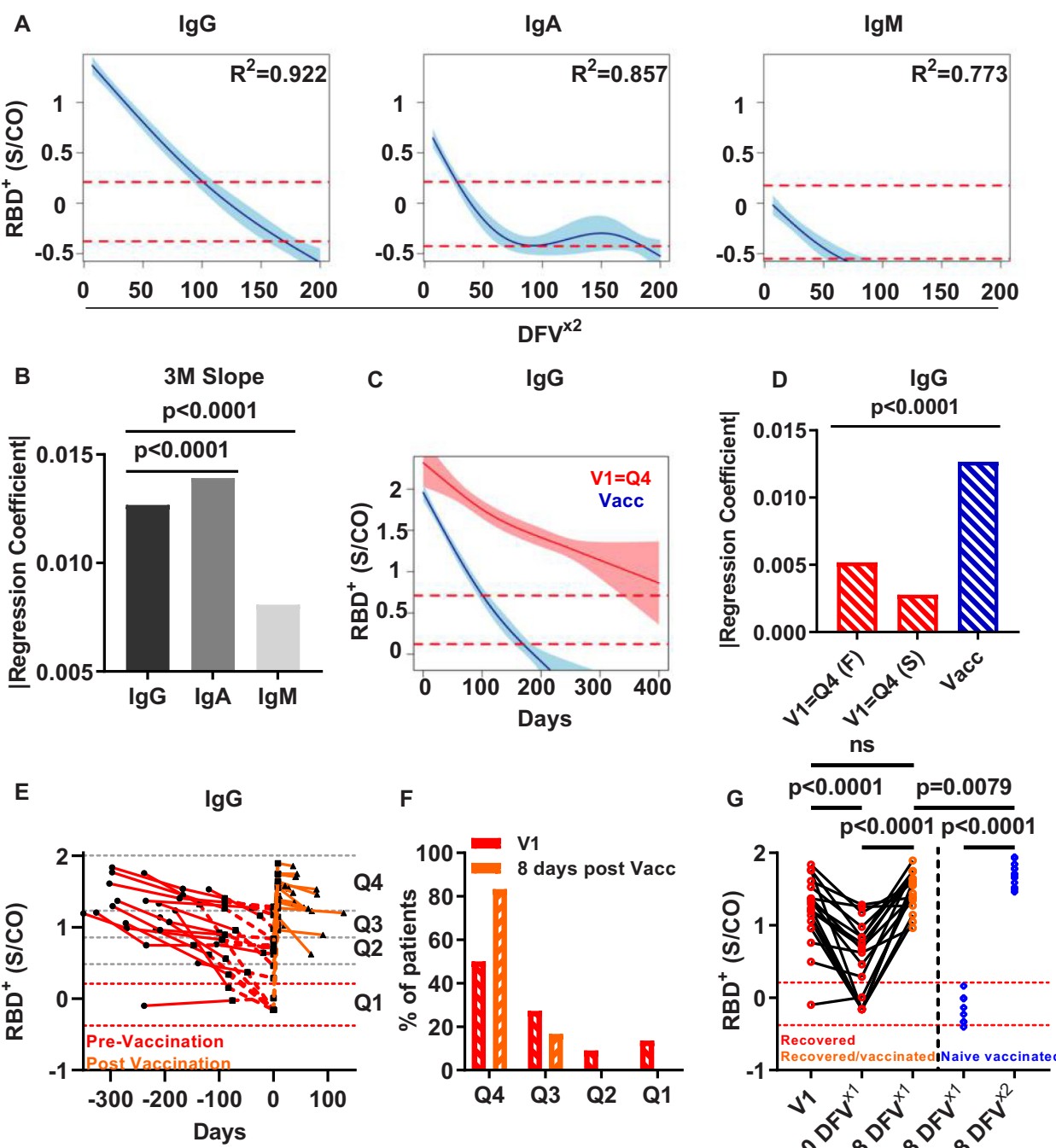

**Fig 6. RBD+ antibody longitudinal kinetics in naïve vaccinees and recovered/vaccinated patients.** (A) RBD+ IgG/A/M longitudinal kinetics in naïve vaccinees following two vaccine doses (n = 17). GAMM analysis was applied, and regression coefficient was determined. The 95% CI is shown in turquoise. Y-axis $\log_{10}$ and X-axis show days following vaccination ($DFV^{x2}$). Dashed red lines indicate the upper and lower limit of negative control samples (n = 27). (B) Bar plot comparing the regression coefficient obtained for IgG, IgA, and IgM during the 90 $DFV^{x2}$ (3 months following vaccine, 3M slope). For all plots $R^2$ is indicated, and p value, slopes, and effective degree of freedom (edf) can be found in S2 Table. p values < 0.05 were considered significant. (C) RBD+ IgG longitudinal kinetics in naïve vaccinees in comparison with the kinetics in recovered patients whose IgG levels were assigned to Q4 at V1. Y-axis $\log_{10}$ shows IgG levels (S/CO), x-axis shows DFS for recovered patients (red) and $DFV^{x2}$ for naïve vaccinees (blue). (D) Multiple comparison analysis of the regression coefficients obtained for the fast decay (F) and slow (S) decay rate in recovered patients and naïve vaccinees. Y-axis represents the absolute regression coefficient values. (E) Utilizing each patients' regression model pre-vaccination and the known vaccination day, the RBD+ IgG level was extrapolated to predict the IgG level on the day of vaccination (0 $DFV^{x1}$). IgG levels post vaccination at V4 were used to extrapolate the antibody levels for 8 $DFV^{x1}$. Y-axis represents the RBD+ IgG levels in $\log_{10}$, x-axis represents days in relation to the day of vaccination (i.e. 0 $DFV^{x1}$). Dashed red lines indicate the upper and lower limit of negative control samples

(n = 27). (F) Distribution of the recovered/vaccinated patients among quartiles based on their antibody levels at V1 compared to the extrapolated antibody levels at 8 DFV$^{x1}$. Y-axis shows the percentage of recovered patients that were assigned to each quartile. X-axis shows quartile groups. (G) Comparison of the RBD$^+$ IgG levels in recovered patients at V1, the RBD$^+$ IgG levels at 0 DFV$^{x1}$ and 8 DFV$^{x1}$ in recovered patients, and the RBD$^+$ IgG levels of naïve vaccinees 8 DFV$^{x1}$ and DFV$^{x2}$. Y-axis in log$_{10}$ IgG levels (S/CO), x-axis shows the relative days from vaccination of each sub-cohort. Dashed red lines indicate the upper and lower limits of the antibody levels in the negative control group (n = 27).

(and before vaccination) (Fig 6F). While there was no significant difference between the IgG levels at V1 and IgG levels 8 DFV$^{x1}$ in recovered/vaccinated patients, there was a significant increase in IgG levels 8 DFV$^{x1}$ compared to 0 DFV$^{x1}$. Still, the IgG levels at 8 DFV$^{x1}$ in the recovered/vaccinated sub-group were significantly lower than those in naïve vaccinees at 8 DFV$^{x2}$ (Fig 6G). Nonetheless, the increase in IgG levels at 8 DFV$^{x1}$ in recovered/vaccinated patients was significantly higher than the antibody levels detected 8 DFV$^{x1}$ in naïve vaccinees who had received one dose of the mRNA-vaccine (Fig 6G).

## Discussion

Highlighting the differences between the immune response following natural infection and vaccination may inform us whether the "training" provided by the vaccine effectively stimulates the immune system in such a way that it will provide long-lasting immunity. This important part of the immune response is attributed to the ability of the adaptive immune system to generate immune memory. The maintenance of stable antibody reservoirs can provide a mechanism that aids in mitigating subsequent infections. We thus deemed it important to reveal the longitudinal kinetics of RBD$^+$ antibodies as a means of evaluating the persistence of serological memory in COVID-19 recovered patients. Our 14-month study of the longitudinal kinetics of RBD$^+$ antibodies in COVID-19 recovered patients revealed that all Ig antibody isotypes decayed over the study period. The decay profile of IgG differed from the profiles of IgA and IgM, with IgG showing a two-phase decay compared to an almost linear decay profile of IgA and IgM. The IgG decay rate in the first 200 DFS was the most rapid, followed by a significantly slower decay than that of IgA and IgM. This decay profile suggests that the robust IgG response induced in the active disease phase declines rapidly, and that this decay rate is moderated over time. This rapid decay in IgG titers was also reported at the beginning of the COVID-19 pandemic by Ibarrondo et al., who observed a similar pattern of decay over the 120 days following the onset of symptoms [24]. The two-phase decay profile was also observed in anti-nucleocapsid IgG levels over a period of 7 months [36]. A possible explanation for the two-phase decay may be derived from the kinetics of the antibody-secreting cells (ASC) induced following infection with SARS-CoV-2. Following the immunogenic viral challenge, the immune system will generate a wave of plasmablasts, which are short-lived plasma cells. These cells may generate the high levels of antibodies in proximity to the challenge. A subset of the plasmablasts will migrate to the bone marrow to become long-lived plasma cells that continue to secrete antibodies into the blood circulation [37]. The continuous decay of antibodies suggests that these long-lived plasma cells do not persist over time. Indeed, it was previously reported that SARS-CoV-2-specific long-lived plasma cells were detected in the bone marrow for as long as 6 months (180 days), but that report did not examine their persistence over a longer period, such as that examined in this study [23]. Thus, it is possible that temporal analysis over an extended period of time would have revealed a decrease in frequency of long-lived plasma cells in accordance with the decay in the antibody levels in the circulation.

It was found in this study that antibody levels in proximity to recovery varied among the different patients and hence that the decay rate of the antibodies also varied among the patients. Thus, it was important to examine longitudinal antibody kinetics both at the

individual level and within groups that exhibited similar antibody levels following recovery. To this end, we examined the longitudinal kinetics of RBD$^+$ antibodies in a cohort of 66 COVID-19 recovered patients from whom blood samples had been drawn at several time points (visits). First, the patients were stratified to quartiles by their antibody levels at the first visit (V1). Next, the RBD$^+$ antibody levels of serum samples from the patients in each quartile were used to determine the regression coefficient that represents the decay rate. Notably, since the decay rate profile in several quartile groups and isotypes fitted a two-phase decay regression, a comparison was made between the fast and slow stages of the decay rates based on the linear regression coefficients obtained from each decay stage. It was found that the decay rate is affected by the robustness of the response as determined by the antibody levels at V1. Hence, patients who exhibited higher antibody levels at V1 (i.e., Q4 vs. Q3) experienced a faster decay rate. This observation can largely be seen across all isotypes and quartile groups, with respect to the separate F and S slopes, the only distinction being the Q2 regressions in IgA and IgM, which show a linear decay profile.

Importantly, we also aimed to reveal the differences of antibody kinetics between recovered patients and naïve vaccinees who had received two doses of the BNT162b2 mRNA-vaccine. Naïve vaccinees elicited a robust response, as observed at 8 DFV$^{x2}$, and the IgG levels were assigned to the highest quartile (Q4). However, the IgG decay rate was found to be faster than that in recovered patients, and at an average of 185 DFV$^{x2}$, the antibody levels in all naïve vaccinees had fallen to the threshold of the negative controls. Comparison of the IgG decay rate between COVID-19 recovered patients who exhibited the highest antibody levels (Q4) at V1 and the levels in naïve vaccinees following the second dose revealed a significantly faster decay in the naïve vaccinees. We deemed this to be the most accurate comparison, since the Q4 recovered group exhibited antibody levels at V1 that were similar to those elicited in naïve vaccinees at 8 DFV$^{x2}$. Of note, it is possible that the antibody levels in a larger cohort of naïve vaccinees would demonstrate a highly heterogeneous distribution and hence that not all individuals would present antibody levels that would be assigned to the highest quartile (Q4). Such heterogenicity would have an effect on the decay profile (see limitations of the study).

Finally, we examined the effect of one vaccine dose on the antibody levels in COVID-19 recovered patients. The single vaccine dose did indeed cause a significant increase in the RBD$^+$ IgG levels. As the vaccine was administered, on average, 222 DFS, the vaccination produced a robust response, with IgG titers reaching the same level as that detected in proximity to recovery (V1). Nevertheless, the IgG levels in recovered/vaccinated COVID-19 patients were significantly lower than those in naïve vaccinees following two vaccine doses.

Overall, the data derived from this study provides new insights into the longitudinal kinetics of antibodies 14 months following recovery and highlights the differences in antibody persistence between recovered patients and naïve vaccinees. Based on the results of this study, we posit that the longitudinal antibody kinetics in COVID-19 recovered patients differs from the kinetics in naïve vaccinees due to fundamental differences between the mechanisms involved in the activation of the adaptive arm of the immune response. While natural infection involves a full systemic activation, including the innate immune arm, mRNA vaccination may not induce full systemic activation. In the case of mRNA-vaccines, this deficiency could affect the ability of the immune system to maintain sufficient levels of the long-lived plasma cells that support the maintenance of the antibody reservoir over time. In the case of COVID-19 recovered patients, it was demonstrated by Turner et al. [23] that long-lived plasma cells persist over time, and this may explain the slower decay rate that we detected in the recovered patient cohort. Of note, these considerations were instrumental in the decision-making process of the Israel Government to lead the way to the currently accepted the three-dose vaccination strategy.

## Limitations of the study

Some limitations should be taken into account when evaluating the results of this study. Since this study was performed on a sample cohort of the Israeli population, our data might not fully represent or reflect trends found in populations outside of Israel. In addition, due to the lack of willingness of patients to give blood on their fourth follow-up visit (we had data for only 2.6% of the total cohort or 7.6% of the follow-up cohort), we had only a small number of data points at V4, which is reflected by a wide range in the 95% CI. Similarly, the cohort of SARS-CoV-2-naïve vaccinees recruited was a relatively small—17 vaccinees. (In recruiting volunteers for the naïve cohort, we did not aim to reach an equally large cohort of COVID-19 recovered patients, since they were not the focus of this research, and we were also limited by ethical constraints.) Nevertheless, despite the above limitations, the availability of follow-up samples for the cohort of naïve vaccinees provided us with sufficient statistical power for the study (using a repeated measures mixed model).

An additional limitation may derive from the use of RBD as the probe to measure the antibody levels rather than the full S protein. Antibodies that target other regions of the S protein are indeed elicited in naturally infected and vaccinated individuals. It is possible that measuring anti-S protein antibodies would have affected the regression model presented in the current study. Nevertheless, as RBD$^+$ antibody levels were shown to be correlated with virus neutralization and are considered the most important antibodies in preventing severe COVID-19 disease, we chose to focus on the longitudinal kinetics of these antibodies.

Finally, under the ethical limitations of this study, participants' decisions to donate blood samples and to be vaccinated were made of their own accord and according to the accepted guidelines (such as intervals between vaccination doses) and resources provided at the time of the study by the Israel Ministry of Health (such as the particular vaccine administered to the population).

## Methodology

### Ethics statement

All patients and naïve individuals provided informed written consent for the use of their data and clinical samples for the purposes of the study, and collection of blood samples was performed under institutional review board approvals (Tel Aviv University) numbers 0001281–4 and 0000406–1. Blood samples were collected at the Hasharon Hospital, Rabin Medical Center under ethical approval number 0265–20. A total of 287 blood samples were obtained from 192 recovered patients, all confirmed positive for SARS-CoV-2 by qPCR. Follow-up samples were obtained from 66 COVID-19 recovered patients at approximately 90-day intervals over three or four clinic visits (V1-V4). Additionally, 18 of the 'follow-up' COVID-19 recovered patients received a single dose of the BNT162b2 vaccine. Samples from 17 naïve individuals who received two doses of the BNT162b2 vaccine were collected at four visits following the second vaccine dose at 8, 35, 91, and 182 DFV$^{x2}$. Negative controls comprised sera from 27 healthy individuals that had been collected prior to the COVID-19 pandemic.

COVID-19 recovered participants were classified into levels of disease severity according to the parameters provided by the Israel Ministry of Health (https://www.gov.il/en/departments/general/corona-confirmed-cases). Accordingly, for patients with a positive qPCR for SARS-CoV-2, those exhibiting fever, cough, malaise, and loss of taste and smell were considered as "mild"; those with pneumonia, as "moderate"; and those with a respiratory frequency of more than 30 breaths per minute, 93% oxygen saturation or less as "severe." Any patient exhibiting

more severe parameters, such as a need for ventilation, and/or suffering from multiple organ dysfunction was also assigned to the "severe" category for the purposes of this study. COVID-19 recovered patients were stratified into two groups: one consisted of all "mild" cases, and the other of the "moderate" and "severe" cases.

## Sample collection

All blood samples were collected into BD Vacutainer K2-EDTA collection tubes. Isolation of serum and peripheral blood mononuclear cells (PBMCs) were performed by density gradient centrifugation, using Uni-SepMAXI+ lymphocyte separation tubes (Novamed) according to the manufacturer's protocol.

## Expression and purification of recombinant protein

The plasmid for the expression of recombinant SARS-CoV-2 RBD was kindly provided by Dr. Florian Krammer, Department of Microbiology, Icahn School of Medicine at Mount Sinai, New York, NY, USA. The RBD sequence is based on the genomic sequence of the first virus isolate, Wuhan-Hu-1, which was released on 10 January 2020 [13]. The plasmid for the expression of hACE2 was kindly provided by Dr. Ronit Rosenfeld, Israel Institute for Biological Research (IIBR). The cloned region encodes amino acids 1–740 of hACE2 followed by an 8×His tag and a Strep Tag at the 3' end, cloned in a pCDNA3.1 backbone. Recombinant RBD and hACE2 were produced in Expi293F cells (Thermo Fisher Scientific) by transfection with purified mammalian expression vector, using an ExpiFectamine 293 Transfection Kit (Thermo Fisher Scientific), according to the manufacturer's protocol, as described previously [13]. Supernatants from transfected cells were purified on a HisTrap affinity column (GE Healthcare) using a two-step elution protocol with 5 column volumes of elution buffer supplemented with 50 mM imidazole in PBS, pH 7.4 followed by 250 mM imidazole in PBS, pH 7.4. Elution fractions containing clean recombinant proteins were merged and dialyzed using Amicon Ultra (Mercury) cutoff 10K against PBS (pH 7.4). Dialysis products were analyzed by 12% SDS–PAGE for purity, and concentration was determined using Take-5 (BioTek Instruments). Purified recombinant proteins were biotinylated using the EZ-Link Micro-NHS-PEG4-Biotinylation kit (Thermo Scientific), according to the manufacturer's protocol.

## Assay for serum anti-SARS-CoV-2 RBD antibodies

RBD+ Ig levels in sera were determined using 96-well ELISA plates that were coated overnight at 4˚C with 2 μg/mL RBD in PBS (pH 7.4). After the coating solution was discarded, the ELISA plates were blocked with 300 μL of 3% w/v skim milk in PBS for 1 h at 37˚C. The blocking solution was then discarded, and duplicates of serum diluted 1:300 in 3% w/v skim milk in PBS were added to the microwells. Negative control serum samples were also tested in single wells at the dilution of 1:300, with a range of 8–24 control samples in each plate. ELISA plates were then washed three times with PBST and 50 μL of horseradish peroxidase (HRP) conjugated goat anti-human IgG (Jackson ImmunoResearch, #109035003) / anti-human IgM (Jackson ImmunoResearch, #109035129) / anti-human IgA (Jackson ImmunoResearch, #109035011) secondary antibodies were added to each plate at the detection phase (50 μL, 1,5000 ratio in 3% w/v skim milk in PBS) and incubated for 1 h at room temperature, followed by three washing cycles with 0.05% PBST. Development was carried out by adding 50 μL of 3,3',5,5'-tetramethylbenzidine (TMB) for 5 min of exposure, immediately followed with reaction quenching by adding 0.1 M sulfuric acid. Plates were read using the Epoch Microplate Spectrophotometer ELISA plate reader at wavelengths of 450 and 620 nm. Reads from the 620

nm wavelength were subtracted from those obtained at the 450 nm wavelength, resulting in a final signal value for each well. A total "negative control value" for each plate was computed as the average of all O.D. 450 obtained from negative control samples, plus addition to one standard division value. A "signal over negative" (S/CO) value for each well was then produced. All sera were evaluated in two independent technical repeats, resulting in 4 S/CO values for each serum sample.

## Computational and statistical analysis

**Bootstrapping & quartile establishment.** We stratified the RBD$^+$ antibody levels into quartiles by using the bootstrap method. Using an R code script, 50 randomly selected values from the extended serum sample cohort (n = 287) were used to set quartile thresholds according to the mean values derived from the bootstrap iterations, a process that was then repeated 10,000 times. The produced 10,000 quartile values were then averaged to four final quartile threshold values (S3B Fig). These values were then used to assign each measured antibody level at V1 into its respective quartile range (i.e. from low to high Q1-Q4, respectively). The benefit of bootstrapping data lies in its ability to produce minimal CI inference and minimize the effects of outlier data points in relatively small sample sizes that might not be evenly distributed [38].

**Statistical analysis.** Data analysis for the generation of plots in Figs 3, 4 and 6 was conducted using a GAMM. Correlations between titer results of ELISA technical repeats were analyzed using Pearson's correlation. A Mann-Whitney test was used to compare two independent groups with continuous variables (Figs 2 and 6G). A comparison of fit tests was used to compare regression models (Figs 3, 5 and 6). A Wilcoxon test was used to compare two or more sets of dependent groups (Fig 6G). All reported P values were two-tailed, and p values less than 0.05 were considered statistically significant. All statistics were performed with GraphPad Prism 9.0.2 (GraphPad Software) and R code (packages "mgcv" [39,40] and "lme4" [41]).

## Supporting information

**S1 Table. Demographic characteristics of the study cohort.** **A**. COVID-19 recovered patients **B**. Follow up recovered patients (sub-cohort of the recovered patients). **C**. Follow up recovered patients who received one BNT162b2 mRNA-vaccine dose (recovered/vaccinated). **D**. Naïve vaccinees who received two doses of the BNT162b2 mRNA-vaccine.
(DOCX)

**S2 Table. Generalized additive mixed model (GAMM) analysis results as determined for patient quartile groups (supplementary data for Figs 3–5).** Slopes represent the linear regression coefficients either for the fast (F) or slow (S) phases of the decay or a single slope of a linear profile. R$^2$ and P values are indicated. The effective degrees of freedom (edf) estimated from GAMMs were used as a proxy for the degree of linearity/non-linearity relationships. An edf of 1 is equivalent to a linear relationship, an edf > 1 indicates a non-linear relationship [42].
(DOCX)

**S3 Table. Generalized Additive Mixed Model (GAMM) analysis results as determined in naïve vaccinated group (supplementary data for Fig 6).** Slopes represents the linear regression coefficients found in the first 90 DFV$^{x2}$ of a profile of a linear profile. R$^2$, and P values are indicated. The effective degrees of freedom (edf) estimated from generalized additive mixed models were used as a proxy for the degree of linearity/non-linearity relationships. An edf of 1

is equivalent to a linear relationship, an edf > 1 indicates a non-linear relationship [42].
(DOCX)

**S1 Fig. Characteristics distribution of donors according to visit. (A)** Demographic distribution (sex, age) and COVID-19 severity of recovered patients. The distribution was binned according to patients that contributed at each visit. (B) Demographic distribution of the naïve vaccinees group (n = 17).
(EPS)

**S2 Fig. Scatter plot and linear regression model.** RBD$^+$ IgG (A) IgA (B) and IgM (C) titers as measured in all serum samples of recovered COVID-19 patients (n = 287) are plotted. Y-axis $\log_{10}$ of S/CO and x-axis, days following symptoms (DFS). Linear regression was applied (solid blue line). Dashed red lines represent the maximum and minimum range of the antibody levels as measured in negative control samples (n = 27).
(EPS)

**S3 Fig. Stratification of the heterogeneous RBD$^+$ IgG/A/M levels at V1. (A)** Scatter plots of the heterogeneous distribution of RBD$^+$ antibody levels in serum samples collected at V1. (B) X,Y plot of quartile determination using bootstrapping: RBD$^+$ antibody levels were randomly selected (n = 50) out of the dataset obtained from all recovered patients (n = 287), and quartile thresholds were set according to the mean values derived from $10^4$ iterations. Y-axis in $\log_{10}$ for each bootstrap round and x-axis show the iteration number. (C) Classification of RBD$^+$ antibody levels by quartiles: RBD$^+$ antibody levels in recovered patients at V1 were classified to their respective quartile (i.e. from low to high Q1-Q4, respectively). Y-axis, $\log_{10}$ antibody levels signal/cutoff ratio (S/CO). X-axis shows the quartile groups (color coded); n indicates the number of individuals assigned to each quartile based on their antibody levels at V1.
(EPS)

**S4 Fig. Regression coefficients of the longitudinal kinetics of RBD$^+$ IgG/A/M by quartiles.** A line graph representing the regression coefficients derived from a GAMM analysis as determined in Fig 4. Y-axis in $\log_{10}$ and x-axis, days following symptoms (DFS). Dashed red lines represent the maximum and minimum range of the antibody levels as measured in negative control samples (n = 27). Regression models that were found to be non-linear (two-phase) were divided into two linear regression phases. The change of the linear regression profile (phase) was estimated to be at 200 DFS for IgG and IgM and 250 DFS for IgA.
(EPS)

**S5 Fig. Scatter plot of RBD$^+$ IgG levels in vaccinated naïve individuals over time.** Each dot represents the RBD$^+$ IgG/A/M levels at a single time point in accordance with the assigned quartile. The data points were used to generate the regression model as shown in Fig 6A. Y-axis $\log_{10}$ IgG/A/M levels (S/CO). x-axis represents days following two doses of vaccination (DFV$^{x2}$). Dashed grey lines represent quartile thresholds. Dashed red lines indicate the upper and lower limits of the antibody levels in the negative control group (n = 27).
(EPS)

**S6 Fig. Extrapolation of longitudinal data points based on a regression model.** Extrapolated data points for the antibody levels at 0 DFV$^{x1}$ were based on the regression model derived from the antibody kinetics as determined from follow-up visits for each individual. (A) Recovered/vaccinated COVID-19 patient number S106 donated blood at 4 visits. V1–V3 took place before the patient was vaccinated. Days following onset of symptoms for V1–V3 are shown in the table, as are the measured RBD$^+$ IgG titers. The patient was vaccinated at 366 DFS, and that day was also designated 0 days following vaccination (DFV$^{x1}$); a blood sample was not

obtained at that time or at 8 DFV$^{x1}$. V4 for patient S106 was 41 DFV$^{x1}$, at which visit a blood sample was obtained. Question marks in the table indicate that no information was available for the IgG titers at those time points. Based on the available data points, the description of the IgG kinetics is presented in the graph. (B) Titers for 0 DFV$^{x1}$ and 8 DFV$^{x1}$ were extrapolated based on the regression model generated on the basis of the visits prior to the vaccination. The 0 DFS data point was extrapolated based on the regression model for the second (slow) phase of the decay (i.e. based on V2 and V3) and the 8 DFS data point was extrapolated based on the regression model in the first (fast) phase of the decay and the data point generated from the serum sample obtained at V4 (i.e. V1 and V2 and V4). The modified graph that is based on the actual and extrapolated data points is shown.
(EPS)

**S1 Data.** -**The numerical data used in all figures.**
(ZIP)

## Acknowledgments

We wish to thank Prof. Itai Benhar and Dr. Limor Nahary from the Shmunis school of Biomedicine and cancer research, Tel Aviv University, for their assistance in providing the initial RBD and hACE2 reagents, Dr. Ronit Rosenfeld from the Israel Institute for Biological Research (IIBR) for her generous assistance and Prof. Mordechai Gerlic and Prof. Ariel Munitz from the Sackler Faculty of Medicine, Tel Aviv University. We would also wish to thank Dr. Keren Agay-Shay from the Bar-Ilan University Azrieli Faculty of Medicine, Safed and Dr. Ofir Levy from the school of zoology at Tel Aviv University for their helpful insights regarding the statistical analysis.

## Author Contributions

**Conceptualization:** Tsuf Eyran, Anna Vaisman-Mentesh, David Taussig, Ran Tur-Kaspa, Dana Marcoviciu, Dror Dicker, Yariv Wine.

**Data curation:** Tsuf Eyran, Anna Vaisman-Mentesh, David Taussig, Dana Marcoviciu.

**Formal analysis:** Tsuf Eyran, Anna Vaisman-Mentesh, David Taussig, Ligal Aizik, Dror Dicker, Yariv Wine.

**Funding acquisition:** Yariv Wine.

**Investigation:** Tsuf Eyran, Anna Vaisman-Mentesh, David Taussig, Dror Dicker, Yariv Wine.

**Methodology:** Tsuf Eyran, Anna Vaisman-Mentesh, David Taussig, Yael Dror, Ligal Aizik, Aya Kigel, Shai Rosenstein, Yael Bahar, Dor Ini, Ran Tur-Kaspa, Tatyana Kournos, Dana Marcoviciu, Dror Dicker, Yariv Wine.

**Project administration:** Yael Dror, Dana Marcoviciu, Dror Dicker, Yariv Wine.

**Resources:** Dror Dicker, Yariv Wine.

**Software:** David Taussig, Ligal Aizik.

**Supervision:** Dror Dicker, Yariv Wine.

**Validation:** Tsuf Eyran, Anna Vaisman-Mentesh, David Taussig.

**Visualization:** Tsuf Eyran, David Taussig, Ligal Aizik, Yariv Wine.

**Writing – original draft:** Tsuf Eyran, Dror Dicker, Yariv Wine.

**Writing – review & editing:** Tsuf Eyran, David Taussig, Dror Dicker, Yariv Wine.

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
