## [Decision Letter · Decision Letter 0]

10 Nov 2021

Dear Dr. Wine,

Thank you very much for submitting your manuscript "The longitudinal kinetics of antibodies in COVID-19 recovered patients over 14 months" for consideration at PLOS Pathogens. As with all papers reviewed by the journal, your manuscript was reviewed by members of the editorial board and by several independent reviewers. In light of the reviews (below this email), we would like to invite the resubmission of a significantly-revised version that takes into account the reviewers' comments.

We cannot make any decision about publication until we have seen the revised manuscript and your response to the reviewers' comments. Your revised manuscript is also likely to be sent to reviewers for further evaluation.

Sincerely,

Florian Krammer, PhD

Associate Editor

PLOS Pathogens

Adolfo García-Sastre

Section Editor

PLOS Pathogens

Kasturi Haldar

Editor-in-Chief

PLOS Pathogens

orcid.org/0000-0001-5065-158X

Michael Malim

Editor-in-Chief

PLOS Pathogens

orcid.org/0000-0002-7699-2064

Reviewer's Responses to Questions

**Part I - Summary**

Reviewer #1: In this paper, Eyran and colleagues investigated the RBD-specific antibody kinetics in a cohort of 200 recovered patients (some of which were also vaccinated). They found that RBD-specific antibody levels decreased faster for those antibodies that were associated with the robustness of the response thus, recovered patients that exhibit elevated antibody levels at the first visit, experience faster decay. The authors also studied the longitudinal kinetics differences between recovered patients and naïve BNT162b2 vaccinees and found a significantly faster decay in naïve vaccinees compared to recovered patients.

This study investigates on a relatively large scale vaccine dynamics over time and is therefore of interest in shaping health decisions regarding booster vaccinations. However, data presentation, data interpretation, and embedding in existing literature need to be improved.

Reviewer #2: The manuscript by Dr. Eyran et al. describes the longitudinal kinetics of SARS-CoV-2 antibodies post-infection and vaccination. The manuscript has interesting data and would benefit from additional grammatical editing as there are numerous errors throughout. There are also major concerns with the statistical methods, loss-to-follow up and small sample sizes.

Reviewer #3: The key aim of the manuscript from Eyran and colleagues is to describe the longitudinal kinetics of RBD+ antibodies in recovered COVID-19 patients over the course of 14 months, with or without two-dose vaccination with Pfizer/BioNTech BNT162b2, and how this compares to antibody persistence in n=17 highest titer (Q4) naïve vaccinees. In broad outline, the authors present data in four figures to establish the following conclusions:

° Their assay for the measurement of antibody levels exhibits high reproducibility.

° RBD-directed antibody levels within the cohort differ between males and females for IgG and IgA; higher levels are associated with disease severity; and antibody levels decay over the course of 14 months.

° The rate of temporal antibody decay can be associated with the “robustness” (titer) of the early response (i.e., the decay rate is significantly faster for those patients in whom the antibody levels at the earliest time point (visit 1 “V1”) ranked in the highest quartile (Q4).

° Antibody levels in recovered patients decay more slowly compared to naïve vaccinees.

Much of the data presented in the manuscript agrees with prior published reports and is not novel; however, the computationally defined observation that antibody decay rates may differ between naturally infected vs. vaccinated individuals is potentially novel.

**Part II – Major Issues: Key Experiments Required for Acceptance**

Reviewer #1: I feel like it’s not super clear to what the current work differs from previous work. Can the authors explain this more extensively in intro and discussion to sufficiently differentiate themselves from previous work – this would be especially important for those scientists who are not covid experts (like myself)

Bottom of page 11

”As recovered patients received a vaccine at various DFS, we extrapolated the antibody levels to the day that the patients received the vaccine using the regression formula. The antibody levels at 8 DFV were extrapolated as well by using the regression formulas obtained from naïve vaccinees (Figure 4e)”

Is this really valid? If yes, I’m not sure I follow. Antibody levels at 8 DFV can only be predicted like that if the regression is the same in recovered vaccinees and naive vaccinees, right? Why is there no regression analysis of recovered vaccinees? Please explain

Fig. 3b and c in the IgG column no real difference in regression discernible by eye yet p<0.0001 due to log scale? Please discuss. Also, add data points in Fig 3c. And change red/green to a colorblind safe color scheme.

Reviewer #2: 1) The nonlinear regression models in table 2 do not appear to fit the data particularly well. It looks as though there is a fairly rapid decline that then levels out, but the models do not reflect that. I suggest trying a Gaussian generalized additive mixed model. This model could then also be used to look at decay based on starting antibody quartiles.

2) There was substantial loss to follow-up over the 4 time points in the post-infected individuals with only 12.2% presenting for the 4th draw. It is highly likely that these individuals are not representative of the general population. At a minimum basic information on the characteristics (age, sex, severity) of individuals that contributed at each time point should be presented along with test statistics to evaluate differences. Further, then stratifying by baseline antibody levels leaves very small sample sizes at the later timepoints.

3) The vaccine cohort is extremely small, with only 17 individuals, limiting it’s usefulness.

4) It is not clear how the antibody levels were “extrapolated” to day 0 and day 8. Given the small sample size, it would seem to be very error prone.

Reviewer #3: Major Concerns:

I have no requests for any substantial revisions; the manuscript is well-written.

The fact that only RBD-directed antibodies were studied should be clarified in the Title.

The fact that the differences in antibody kinetics between recovered COVID-19 patients and naïve vaccinees (n=17) was analyzed only for the highest quartile (Q4 highest titer) at first visit (V1) recovered patients — this deserves further clarification in the Discussion.

Given the observation that antibody decay rates in recovered patients correlate with quartile titer at V1, such a quartile-dependency in decay rate may be true for naïve vaccinees, too. In this regard, the analysis by Eyran et al. could be enhanced by the inclusion of more naïve vaccinees, especially those whose vaccine-elicited titers reach only Q3, Q2 or Q1. (I am not asking for additional data [unless it happens to exist] but feel that this limitation of the study deserves more airplay in the Discussion.)

The fact that this study lacks analysis of antibody responses directed to the spike ectodomain, and has focused exclusively on the RBD, should be discussed. For instance, the antibody decay rate of S-directed titers between recovered patients vs naïve vaccinees might be statistically indistinguishable, thus diminishing the key observation reported in the current manuscript (namely, a difference in antibody persistence of RBD-directed IgG between these two groups).

**Part III – Minor Issues: Editorial and Data Presentation Modifications**

Reviewer #1: Low resolution of most figures, please improve figure resolution

Fig 1:

a: Figure does not include the vaccine patients (and depiction of DFV time points)

b: gender → sex (change also in the text)

B: explain MI/SE in figure or just spell it out

B: the legend inside the age circle excludes the ages 51 and 51. Please correct numbers and sign. Same, in the text on page 7, you write <50 and >50, which excludes 50.

C: is a corr coefficient of 0.4 really signaling high reproducibility? Please discuss. And can you also discuss how this variation may influence the variability of results obtained in the later figures?

Fig 2:

Is it possible to add units to y-axes?

Did you consider multiple-testing correction for your analyses?

C: can you show the data points?

Is 17 individuals in the naive vaccinated and 20 in recovered+vaccinated cohort enough? Please discuss.

Confusingly worded bottom of page 11:

“Based on the calculated temporal kinetics we found that approximately 80% of recovered vaccinees experienced an increase in IgG levels reaching the highest quartile (Q4) as opposed to 50% of them being designated to Q4 following recovery (Figure 4f)”

Would like to know:

What about different vaccination intervals (e.g., 6 weeks instead of 3 in between shots)

What about 2*astrazeneca and astrazeneca + mRNA booster and single-shot vaccines?

Reviewer #2: 1) Results are presented in the introduction.

2) Some of the figures are hard to see.

Reviewer #3: Minor Concerns:

There is no callout of Fig. 3c in the caption.

It is unclear how exactly the statistical tests were applied in Fig. 3c

The first paragraph of the Discussion is dispensable and could be deleted.

PLOS authors have the option to publish the peer review history of their article (what does this mean?). If published, this will include your full peer review and any attached files.

Reviewer #1: No

Reviewer #2: No

Reviewer #3: No
---

## [Decision Letter · Decision Letter 1]

31 Mar 2022

Dear Dr. Wine,

Thank you very much for submitting your manuscript "Longitudinal kinetics of RBD+ antibodies in COVID-19 recovered patients over 14 months" for consideration at PLOS Pathogens. As with all papers reviewed by the journal, your manuscript was reviewed by members of the editorial board and by several independent reviewers. While two of the reviewers found that their concerns had been addressed, one of the reviewers still had a major concern.  In light of the reviews (below this email), we would like to invite the resubmission of a significantly-revised version that takes into account the reviewers' comments.

We cannot make any decision about publication until we have seen the revised manuscript and your response to the reviewers' comments. Your revised manuscript is also likely to be sent to reviewers for further evaluation.

Sincerely,

Florian Krammer, PhD

Associate Editor

PLOS Pathogens

Adolfo García-Sastre

Section Editor

PLOS Pathogens

Kasturi Haldar

Editor-in-Chief

PLOS Pathogens

orcid.org/0000-0001-5065-158X

Michael Malim

Editor-in-Chief

PLOS Pathogens

orcid.org/0000-0002-7699-2064

Reviewer's Responses to Questions

**Part I - Summary**

Reviewer #1: The authors have addressed all my concerns.

Reviewer #2: Overall, the authors have responded to my concerns. However, they failed to respond to a particular concern about bias and their response is concerning.

Reviewer #3: The authors have substantially improved the manuscript. They more clearly describe their statistical analysis by the inclusion of a new supplemental figure (Fig. S5) and accompanying main text (e.g., lines 248–277). The authors have also crafted a new and scientifically sound subsection to their Discussion (namely, “Limitations of the study,” lines 363–387) which is much to their credit and appropriately tempers the breadth of their conclusions while also underscoring the novelty of their data set and its analysis. All figures have been improved for both resolution, layout and content.

My requests and concerns have been satisfactorily addressed.

**Part II – Major Issues: Key Experiments Required for Acceptance**

Reviewer #1: The authors have addressed all my concerns.

Reviewer #2: The comment and response in question is:

Reviewer's comment: "At a minimum basic information on the characteristics (age, sex, severity) of individuals that contributed at each time point should be presented along with test statistics to evaluate differences.

" Response: The reviewer's request for an addition of presented information on the characteristics (age, sex, severity) of individuals who contributed at each time point is very much understandable. However, we found this kind of presentation sub-optimal and disproportional to the facts. This is the sole reason that while aiming for a 3-month interval between blood sampling, in practice, we had no executive control of the matter (participants would arrive on their own accord). This results in a non-uniform sampling time frame between participants.

While I completely understand that non-uniform sampling time occurs in observational studies. The authors in question have binned the sampling times and thus there is no reason that the information on who contributed samples at which time points cannot be presented at least in the supplement. They should just bin using the same time points. Their reluctance to provide basic data on their samples and the minimum data necessary to evaluate potential bias in this study due to loss-to-follow up is very concerning.

Reviewer #3: (No Response)

**Part III – Minor Issues: Editorial and Data Presentation Modifications**

Reviewer #1: The authors have addressed all my concerns.

Reviewer #2: (No Response)

Reviewer #3: (No Response)

PLOS authors have the option to publish the peer review history of their article (what does this mean?). If published, this will include your full peer review and any attached files.

Reviewer #1: No

Reviewer #2: No

Reviewer #3: No
---

## [Editor Report · Decision Letter 2]

3 May 2022

Dear Dr. Wine,

We are pleased to inform you that your manuscript 'Longitudinal kinetics of RBD+ antibodies in COVID-19 recovered patients over 14 months' has been provisionally accepted for publication in PLOS Pathogens.

Best regards,

Florian Krammer, PhD

Associate Editor

PLOS Pathogens

Adolfo García-Sastre

Section Editor

PLOS Pathogens

Kasturi Haldar

Editor-in-Chief

PLOS Pathogens

orcid.org/0000-0001-5065-158X

Michael Malim

Editor-in-Chief

PLOS Pathogens

orcid.org/0000-0002-7699-2064
---

## [Editor Report · Acceptance letter]

30 May 2022

Dear Dr. Wine,

We are delighted to inform you that your manuscript, "Longitudinal kinetics of RBD+ antibodies in COVID-19 recovered patients over 14 months," has been formally accepted for publication in PLOS Pathogens.

Best regards,

Kasturi Haldar

Editor-in-Chief

PLOS Pathogens

orcid.org/0000-0001-5065-158X

Michael Malim

Editor-in-Chief

PLOS Pathogens

orcid.org/0000-0002-7699-2064